# Identifying substitutional oxygen as a prolific point defect in monolayer transition metal dichalcogenides

Sara Barja[1,2,3,4,13], Sivan Refaely-Abramson[1,5,12,13], Bruno Schuler [1,13], Diana Y. Qiu[5,6,13], Artem Pulkin[7], Sebastian Wickenburg[1], Hyejin Ryu[8,9], Miguel M. Ugeda[2,3,4], Christoph Kastl[1], Christopher Chen[1], Choongyu Hwang[10], Adam Schwartzberg[1], Shaul Aloni[1], Sung-Kwan Mo [8], D. Frank Ogletree [1], Michael F. Crommie [5,11], Oleg V. Yazyev [7], Steven G. Louie [5,6], Jeffrey B. Neaton[1,5,11] & Alexander Weber-Bargioni[1]

Chalcogen vacancies are generally considered to be the most common point defects in transition metal dichalcogenide (TMD) semiconductors because of their low formation energy in vacuum and their frequent observation in transmission electron microscopy studies. Consequently, unexpected optical, transport, and catalytic properties in 2D-TMDs have been attributed to in-gap states associated with chalcogen vacancies, even in the absence of direct experimental evidence. Here, we combine low-temperature non-contact atomic force microscopy, scanning tunneling microscopy and spectroscopy, and state-of-the-art ab initio density functional theory and GW calculations to determine both the atomic structure and electronic properties of an abundant chalcogen-site point defect common to $MoSe_2$ and $WS_2$ monolayers grown by molecular beam epitaxy and chemical vapor deposition, respectively. Surprisingly, we observe no in-gap states. Our results strongly suggest that the common chalcogen defects in the described 2D-TMD semiconductors, measured in vacuum environment after gentle annealing, are oxygen substitutional defects, rather than vacancies.

[1] Molecular Foundry, Lawrence Berkeley National Laboratory, Berkeley, CA 94720, USA. [2] Departamento de Física de Materiales, Centro de Física de Materiales, University of the Basque Country UPV/EHU-CSIC, Donostia-San Sebastián 20018, Spain. [3] IKERBASQUE, Basque Foundation for Science, Bilbao 48013, Spain. [4] Donostia International Physics Center, Donostia-San Sebastián 20018, Spain. [5] Department of Physics, University of California at Berkeley, Berkeley, Berkeley, CA 94720, USA. [6] Materials Sciences Division, Lawrence Berkeley National Laboratory, Berkeley, CA 94720, USA. [7] Institute of Physics, Ecole Polytechnique Fédérale de Lausanne (EPFL), CH-1015 Lausanne, Switzerland. [8] Advanced Light Source, Lawrence Berkeley National Laboratory, Berkeley, CA 94720, USA. [9] Center for Spintronics, Korea Institute of Science and Technology, Seoul 02792, Korea. [10] Department of Physics, Pusan National University, Busan 46241, Korea. [11] Kavli Energy NanoSciences Institute at the University of California Berkeley and the Lawrence Berkeley National Laboratory, Berkeley, Berkeley, CA 94720, USA. [12]Present address: Department of Materials and Interfaces, Weizmann Institute of Science, Rehovot 7610001, Israel. [13]These authors contributed equally: Sara Barja, Sivan Refaely-Abramson, Bruno Schuler, Diana Y. Qiu. Correspondence and requests for materials should be addressed to S.B. (email: sara.barja@ehu.eus) or to S.G.L. (email: sglouie@berkeley.edu) or to J.B.N. (email: jbneaton@lbl.gov) or to A.W.-B. (email: afweber-bargioni@lbl.gov)

Crystal defects are known to modify semiconductor functionality and are expected to have particularly strong impact on the properties of two-dimensional (2D) materials, where screening is reduced compared to bulk systems[1]. In particular, 2D transition metal dichalcogenides (TMDs) can feature a variety of different defect geometries and related electronic states[2,3]. Consequently, correlating individual structural defects with electronic properties is key for understanding the behavior of and, ultimately, engineering functional 2D-TMDs. However, the experimental identification of individual defects and the direct correlation of these measurements to their electronic structure still remains a challenge.

Chalcogen vacancies are considered to be the most abundant point defects in 2D-TMD semiconductors, and they are theoretically predicted to introduce deep in-gap states (IGS)[4–10]. As a result, important features in the experimental transport characteristic[9], optical response[5,8,10–13] and catalytic activity[14–17] of 2D-TMDs have typically been attributed to chalcogen vacancies, based on indirect support from images acquired by transmission electron microscopy (TEM)[6,9,11,14,18,19] and scanning tunneling microscopy (STM)[13,16,17,20–22]. However, TEM measurements do not provide direct access to the electronic structure of individual defects. The difficulties in discriminating from the native point defects and those created by TEM due to radiation damage effects has been widely reported[3,23]. Furthermore, light substitutional atoms, such as oxygen, will produce only very weak TEM contrast and could be mistaken for vacancies[24]. Altogether, these challenges limit the direct correlation of TEM studies on TMD materials with their macroscopic response, and the optimization of the material's performance if based on these results. Non-invasive STM has been used to study both the structure and the electronic properties of point defects in 2D-TMDs. While scanning tunneling spectroscopy (STS) is a direct probe of the local electronic structure of the individual defects, the interpretation of their chemical nature from atomically resolved STM images in 2D-TMDs is ambiguous[25] due to the convolution of geometric and electronic structure, which is particularly complex for semiconductors. Prominent features in previous STM images are commonly attributed to chalcogen atom positions[25,26]. Presumably, apparent depressions in the STM images of vacancies have been assigned to chalcogen vacancies in 2D-MoS$_2$[16,20–22] and 2D-TiSe$_2$[27,28], while reported as W vacancies in WSe$_2$ samples[13], guided by the absence of IGSs in STS measurements. The strong dependence of the tunneling conditions on the STM contrast of the atomic lattice and the unclear differentiation between chalcogen and metal sublattices in former STM studies has led to a non-consistent interpretation of the defect type across the current literature[13,14,16,17,20–22,27,28].

Sulphur vacancies, and their corresponding IGS, have been held responsible for unexpected catalytic activity in hydrogen evolution reactions reported in MoS$_2$[14,15,17]. This assumption is challenged, however, by the enhanced catalytic activity reported after long-term ambient exposure of MoS$_2$ samples[16], since this enhanced catalysis is attributed to oxygen substitution of S atoms, which have been predicted to lack any IGS[19,29,30]. In order to achieve a fundamental understanding of the effect of defects on the electronic structure, a direct correlation between the atomic and electronic structure of individual defects in 2D-TMDs is required.

In this work, we combine complementary techniques that allow access to the material at the atomic-scale—low-temperature non-contact atomic force microscopy (nc-AFM), STM and STS—together with parallel state-of-the-art first-principles ground- and excited-state calculations using DFT and many-body perturbation theory within the GW approach, respectively, to enable a comprehensive interrogation of the system. We demonstrate how the combination of these methods can reveal the structure of the most abundant type of defects in our 2D-MoSe$_2$ and 2D-WS$_2$ samples. We directly relate atomic and electronic structure through combined nc-AFM and STM/STS measurements of individual point defects in monolayer MoSe$_2$ grown by molecular beam epitaxy (MBE) and in monolayer WS$_2$ grown by chemical vapor deposition (CVD) (see Methods). Although our nc-AFM and STM images of chalcogen defects appear to be consistent with vacancies, a comparison with our DFT and GW calculations establishes these defects as substitutional oxygen at chalcogen sites, consistent with the lack of IGS in the bandstructure. Our comprehensive joint experimental and theoretical study reveals substitutional oxygen as a prolific point defect in 2D-TMDs and provides critical insight for future defect engineering in these systems.

## Results

### Atomic structure of point defects in 2D-MoSe$_2$ and 2D-WS$_2$.
Large-scale STM images measured on single layer of MoSe$_2$ and WS$_2$ show predominantly two types of point-defect structures (see Supplementary Fig. 1a). Figure 1a shows a nc-AFM image of the most abundant types of point defects imaged in our 2D-MoSe$_2$ samples, measured using a CO-functionalized tip for enhanced spatial resolution[31]. Based on the known contrast mechanism of CO-tip nc-AFM[31–33] we assign the hexagonal lattice of bright features (higher frequency shifts) to the outer chalcogen atoms, which are close enough to the tip to generate repulsive forces, and the dark features (lower frequency shifts) to the lower-lying metal atoms, whose larger distance from the tip resulted in purely attractive forces. This assignment unambiguously identifies the lattice sites of MoSe$_2$[34], outlined in Fig. 1a. Accordingly, the two main defect features we observe are located on Se-sublattice sites. Figure 1b and c show the STM constant current images of the exact same pair of defects as in Fig. 1a in the valence band (VB; $V_{sample} = -1.55$ V) and conduction band (CB; $V_{sample} = +0.7$ V), respectively. The defect-induced modification of the local density of states (LDOS) for both the VB and CB shows a distinct three-fold symmetry with a spatial extent of about 2 nm. States associated with the corresponding defects in 2D-WS$_2$ measured at energies around the CB onset exhibit an apparent six-fold symmetry (see Supplementary Fig. 1). States with six-fold symmetry are also observed for the equivalent defects in 2D-MoSe$_2$ at a sample bias of about 200 mV above the onset of the CB energy (see Supplementary Fig. 2b and d).

### Electronic structure of point defects in 2D-MoSe$_2$ and 2D-WS$_2$.
Structural defects are well-known to alter the electronic structure of a semiconductor via additional localized states. We study the LDOS of the same type of point defects in MoSe$_2$ and WS$_2$ using STM d$I$/d$V$ spectroscopy. Figure 2a shows a representative d$I$/d$V$ spectrum (red line) measured on top of the left point defect in Fig. 1. The spectrum reveals a well-defined bandgap—equivalent to the gap measured on pristine 2D-MoSe$_2$[25,34] at a location next to the defect (black line)—and lacks IGS. An additional defect state is visible about 300 mV below the VB edge. d$I$/d$V$ spectra measured on the right point defect in Fig. 1 also lacks IGS and exhibits the same qualitative spectroscopic features as the d$I$/d$V$ spectra measured on the left defect (see Supplementary Fig. 2a). d$I$/d$V$ spectra for analogous point defects in 2D-WS$_2$ (blue curve in Fig. 2b) show similar characteristics. The quasiparticle gap near the defect is the same as that measured on pristine 2D-WS$_2$ (black curve in Fig. 2b), and the defect d$I$/d$V$ spectrum exhibits a characteristic defect state below the valence band, similar to the defect in MoSe$_2$. The inset in Fig. 2b shows a spatially resolved conductance scan, d$I$/d$V$ ($x,V$), along the line crossing the point

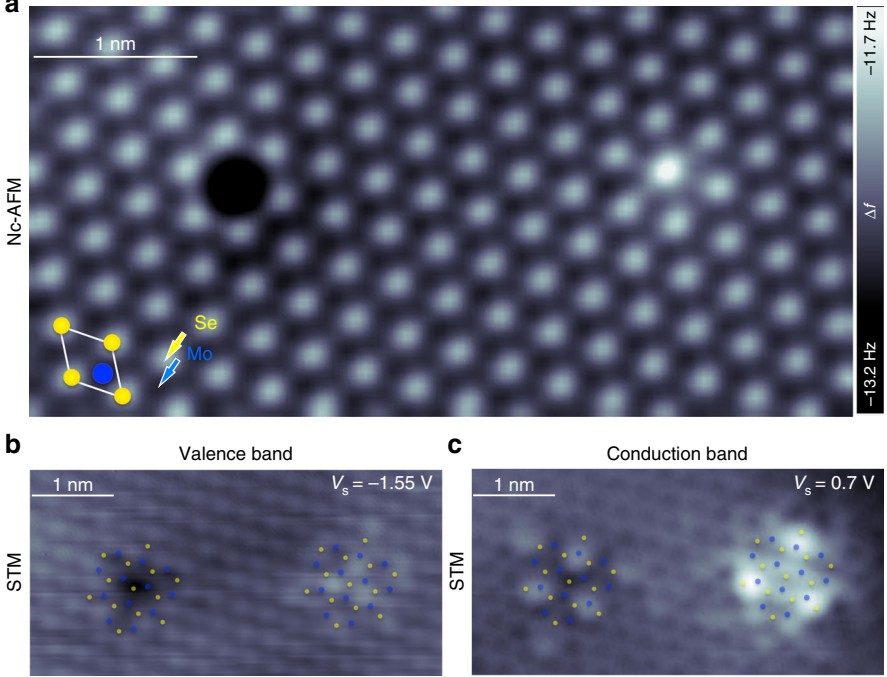

**Fig. 1** Nc-AFM and STM images of the $O_{Se}$ top and bottom defects in 2D-MoSe$_2$. **a** CO-tip nc-AFM image of $O_{Se}$ top in the top Se layer of 2D-MoSe$_2$ (left) and $O_{Se}$ bottom in the lower Se layer facing the graphene substrate. Atomic resolution STM constant current images on the same area as in **a**, measured at the **b** valence and **c** conduction bands edges. Se (yellow dots) and Mo (blue dots) locations are indicated in the images

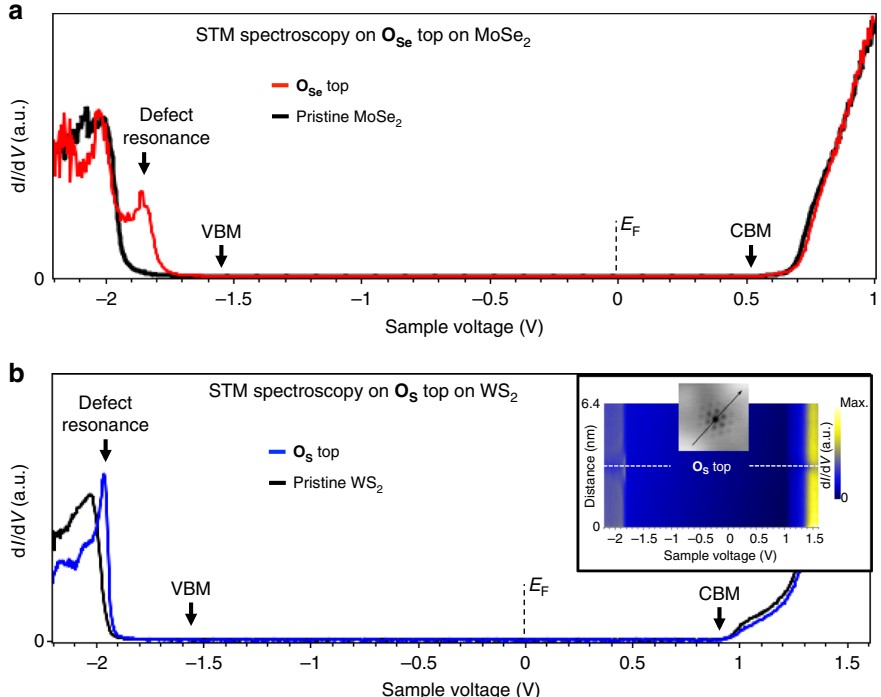

**Fig. 2** Scanning tunnelling spectroscopy of substitutional oxygen in chalcogen site on 2D-MoSe$_2$ and 2D-WS$_2$. **a** Representative STM d$I$/d$V$ spectra acquired on the left defect in Fig. 1—substitutional oxygen at a Se site, $O_{Se}$, in 2D-MoSe$_2$ (red line) do not show deep in-gap states and a badgap equivalent to that measured on pristine sites (black line). Valence band maximum (VBM) and conduction band minimum (CBM) are marked with arrows. An additional defect resonance about 300 mV bellow the VBM it is observed in the defect's spectra. **b** STM d$I$/d$V$ spectra acquired on an substitutional oxygen, $O_S$, at a S site in 2D-WS$_2$ (blue line) also show an equivalent bandgap to that measured on pristine sites (black line), lack of deep in-gap states and a defect resonance deep inside the valence band. Inset: Spatially resolved d$I$/d$V$ conductance scan across the $O_S$ defect depicted in the inset reveals spatially distribution of the defects feature over 2 nm from the center of the $O_S$ (dotted-white line). Sample voltage of 0 V represents the Fermi level ($E_F$)

defect in the panel. The line scan reveals the spatial extent of the described spectroscopic features within 2 nm of the center of the defect. The gap edges are determined by taking the logarithm of the $dI/dV$, as described in ref. [25]. The absence of any IGS resonance associated with the defect is in stark contrast to previous expectations for chalcogen vacancies[4–7,10,18] in both MoSe$_2$ and WS$_2$.

To validate our observation, we perform a series of control experiments on both MoSe$_2$ and WS$_2$ to exclude various scenarios that might prevent IGS from being observed by STS. We are able to rule out orthogonality of the wave functions of the tip and defect states leading to a vanishing tunneling matrix element; dynamic and static charging of the defect and the influence of the graphene substrate (see Supplementary Fig. 3 and Supplementary Fig. 5). The control experiments on both MoSe$_2$ and WS$_2$ establish that IGS are not associated with the most abundant type of defects in our TMD samples.

**Theoretical approach and comparison with the experiments.** Since our STS data curves do not feature IGS, we turn to first-principles DFT and GW calculations for further insight. Prior calculations of S vacancies and substitutional point defects[4–7,10,18,19,29,30,35–37] in 2D-MoS$_2$ have shown that energy levels associated with defects can vary significantly, depending on whether the defect is a vacancy or a substituted atom. Based on growth conditions for both single layers (MoSe$_2$ and WS$_2$) and the sample treatment prior to our scanning probe measurements, oxygen, carbon, silicon, nitrogen, and hydrogen may potentially substitute chalcogen atoms. We can exclude S and Se substituents in, respectively, MoSe$_2$ and WS$_2$ samples, as they are grown in different experimental set-ups, avoiding cross contamination of chalcogen sources. Additionally, since S and Se atoms possess similar van der Waals radii they would result in similar nc-AFM images and are not expected to appear as a depression in the chalcogen lattice. According to prior DFT calculations, hydrogen, nitrogen, carbon, and silicon substituents form IGS within the semiconducting gap[10,19] (See also Supplementary Fig. 5). Oxygen substitution, on the other hand, is predicted to suppress the deep in-gap states associated with the vacancies in MoS$_2$[19,29], and oxygen is present during the CVD growth process of WS$_2$. While the MBE samples are Se-capped to prevent air exposure during the transfer to the STM set-up, desorption of atmospheric H$_2$O, O$_2$, or CO$_2$ from the air-exposed sample holder is likely to occur during the annealing process for decapping (See Methods for further description). Recent DFT calculations propose O$_2$ molecules to chemisorb and dissociate on the chalcogen vacancies[30,38], leading to a stable system formed by a O substitutional chalcogen and a O adatom, further supporting the likelihood that O may be incorporated into the MBE samples. Pristine sulfur vacancies can, however, be generated during annealing in-vacuum[10,39], as they present low formation energies[18,19]. By annealing our WS$_2$ samples at about 600° C in ultra high vacuum (UHV) conditions, we observe the formation of sulfur vacancies. The STS spectra of the $V_S$ in WS$_2$ reveals a characteristic fingerprint with two narrow unoccupied defect states[40]. The identification of in-vacuum-generated $V_S$, with a robustly different electronic structure than $O_S$, validates the assignment of the observed defects in Fig. 1 as substitutional oxygen.

Standard DFT calculations are well known to underestimate bandgaps and quasiparticle energy levels[41]. For point defects in particular, DFT can incorrectly predict the relative energies of localized defect and delocalized bulk states[42,43]. Our TMDs feature both defect states, which are localized near and at the defect, and non-defect extended states associated with the pristine

system[5,6,18]. To compute the energies of both defect and extended TMD states with spectroscopic accuracy, we use ab initio many-body perturbation theory within the GW approximation[44,45], correcting standard DFT energies with additional many-electron self-energy effects relevant to charged excitations in these systems. Our GW calculations are expected to accurately predict the energies of both localized and extended states, and has previously been shown to produce accurate bandgaps and electronic structures of pristine TMDs[46–48].

We first relax the atomic coordinates of a monolayer of MoSe$_2$ with several different point defect types, including a Se vacancy and substitutional atoms, using DFT[49] (see Methods). We then use a force-field model to simulate nc-AFM images, following a previously established method by Hapala, et al.[33] Fig. 3a–c show the relaxed atomic structure of a Se vacancy ($V_{Se}$, Fig. 3a), and two different substitutional (oxygen, $O_{Se}$, -Fig. 3b- and hydrogen, $H_{Se}$ – Fig. 3c) Se defects in MoSe$_2$. The corresponding simulated nc-AFM images from these three types of point defects are shown in Fig. 3d–f when the defect is located in the top layer (facing the tip in our experimental system) and in Fig. 3g–i, for which the defect is placed in the bottom layer (facing the underlying graphene layer in experimental system). Interestingly, since both H and O substituents are recessed into the chalcogen layer, the simulated nc-AFM contrast in images from all three types of atomic defects is equally compatible with the measurements in Fig. 1a. The atomic sized depression is assignable to the missing Se atom or the substituent atom in the upper Se-sublattice facing the tip. An apparently protruding Se atom can be identified as the equivalent defect in the bottom Se-sublattice facing the underlying graphene layer; the bottom defect slightly pushes the Se atom upward relative to the pristine lattice. In 2D-WS$_2$ we identified the counterpart defects on the top and bottom S-sublattices, which exhibit the same nc-AFM morphology as described for MoSe$_2$ (see Supplementary Fig. 1).

To identify which defects are likely observed in Fig. 1, we turn to calculations of their quasiparticle electronic structure using ab initio many-body perturbation theory within the GW approximation[44,45] since STS measures the energies of quasiparticle excitations, i.e., addition or removal of an electron. We perform DFT and GW calculations for both the bare Se vacancy and aforementioned substituted Se defects in MoSe$_2$. (Prior calculations of the bare chalcogen vacancy in WS$_2$ suggest that the defect electronic structure is qualitatively similar in both TMDs[40,50].) We emphasize that ab initio GW calculations are crucial to accurately describe the energy levels of both defect and non-defect states, required for comparing our computational results with the experimental observations and defect identification. As shown in Fig. 4a, the calculated GW quasiparticle gap between the valence and conduction band edge states is 2.2 eV. Our GW calculations of the Se vacancy reveal a deep, doubly-degenerate IGS located near the experimental Fermi level (red line), in qualitative agreement with prior DFT results. (Our DFT calculations of the H substitution Se defect also show IGS states, see Supplementary Fig. 5). On the other hand, our GW results for the substitutional O defect show no IGS, which can be rationalized by the fact that O is isoelectronic to Se and S, and the resulting bandgap of 2.1 eV compares well with the experimental gap. The lack of IGS for the substitutional O defect, together with its simulated nc-AFM image, is consistent with the point defect shown in Fig. 1.

We compare the measured spatial distributions of the LDOS around the top $O_{Se}$ defect and the calculated wave functions for both the Se vacancy and the $O_{Se}$ defect. Figure 4b shows a representative $dI/dV$ constant-height conductance map measured at the CB ($V_{sample} = 0.7$ V) of the top $O_{Se}$ defect for MoSe$_2$. The calculated LDOS spatial distribution near the CB edge for both

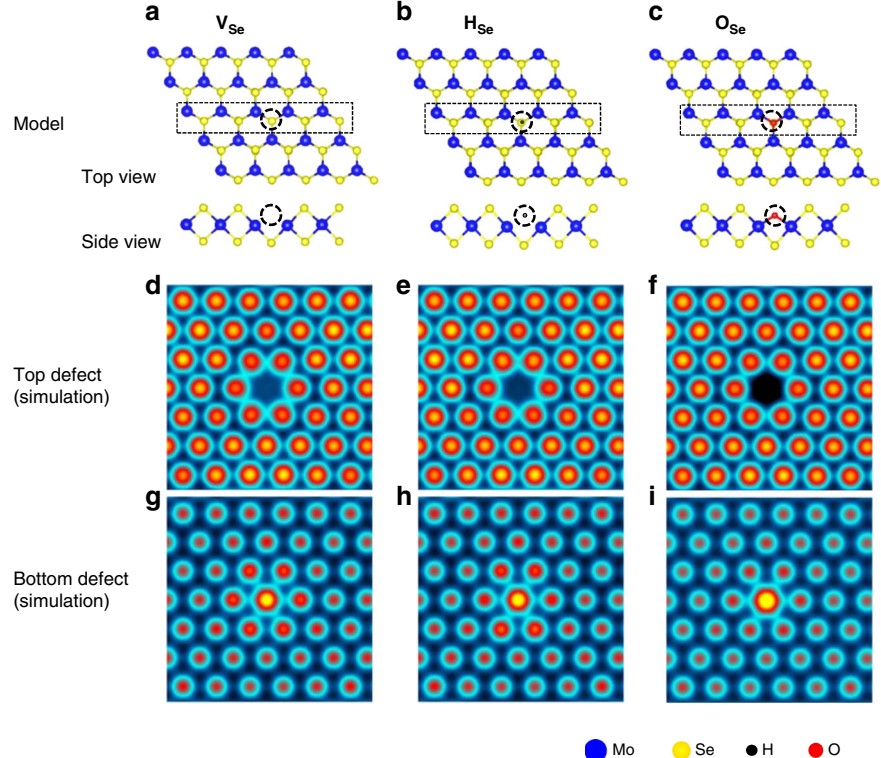

**Fig. 3** Nc-AFM CO-tip simulation of a Se vacancy, H and O substitution. Atomic structure from DFT relaxed coordinates of **a** a Se vacancy ($V_{Se}$), **b** hydrogen substitution ($H_{Se}$), and **c** oxygen substitution ($O_{Se}$) at a chalcogen site in a single layer of MoSe₂. Simulations of the nc-AFM images using a previous established method by Hapala et al.[33] of $V_{Se}$, $H_{Se}$, and $O_{Se}$ placed both **d–f** in the top layer (Se-sublattice facing the tip) and **g–i** in the bottom layer (Se-sublattice facing the underlying graphene layer), respectively

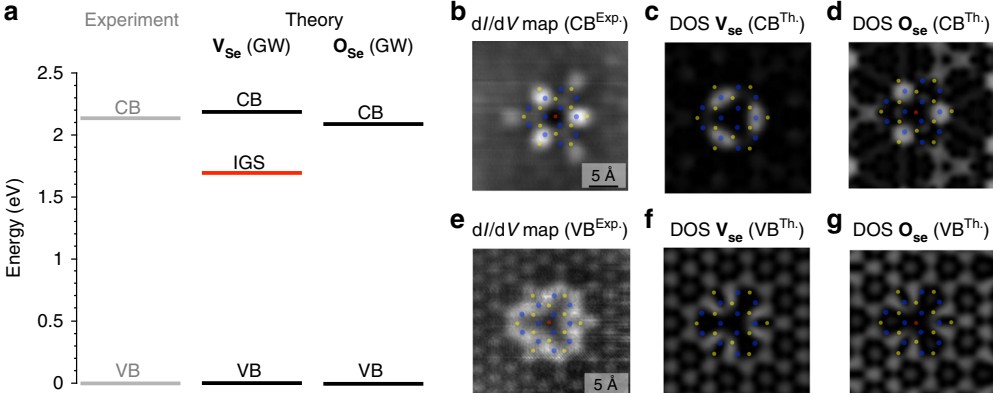

**Fig. 4** Comparison of the band energy diagrams and the local density of states of pristine and O substitution Se defects in 2D-MoSe₂. **a** Band extrema energies extracted from the experimental dI/dV spectra in Fig. 2a (Experiment, gray) are compared to the corresponding energies (Theory, black) calculated using the GW approximation for the bare Se vacancy $-V_{Se}(GW)-$ and a substitutional O at a Se site $-O_{Se}(GW)-$. Energies of valence (VB) and conduction (CB) bands are indicated by black lines; the deep in-gap states (IGS) appearing in the $V_{Se}$ are indicated by the red line, which dictates the Fermi level of the calculated system. To aid comparison, all VB energies have been set to zero. The three-fold symmetry and spatial extent observed in the experimental dI/dV constant-height conductance map measured at the **b** CB energy ($V_{sample} = 0.7$ V) are seen on both **c** the pristine $V_{Se}$ and **d** the $O_{Se}$. Similarly, the experimental spatial extent at the **e** VB ($V_{sample} = -1.5$ V) also reproduced the simulated LDOS of both **f** $V_{Se}$ and **g** $O_{Se}$

the Se vacancy (Fig. 4c) and $O_{Se}$ defect (Fig. 4d) closely resembles the experimental map. Similarly, the simulated LDOS of the vacancy (Fig. 4f) and top $O_{Se}$ defect (Fig. 4g) at the VB edge agree well with experiment, as shown in Fig. 4e. This result reveals limitations of STM imaging for discerning between the two types of defects, as previously addressed in the literature[13,21,22,27,28]. We compare the measured STM image at the CB edge for MoSe₂

with the calculation of the vacancy at both the energy of the deep in-gap state and the CB edge. The simulated STM image for the vacancy at the IGS (see Supplementary Fig. 4) is distinct from the experimental dI/dV maps in shape and registry to the MoSe₂ lattice. We emphasize that access to the position of the Mo sublattice from nc-AFM imaging enables a direct comparison with the calculated wave functions of the defects. This

information is crucial due to the similar symmetries of the $V_{Se}$ IGS wavefunction and the CB wavefunction associated with the $V_{Se}$ and $O_{Se}$ defects.

## Discussion

Taken together, our analysis suggests that the commonly observed point defects in $MoSe_2$ and $WS_2$, growth ex situ by MBE and CVD and measured in UHV after gentle annealing, are O substitutions at Se and S sites, respectively. A number of TEM and STM studies have investigated point defects on various TMD samples[9,18], and the majority of chalcogen-site point defects have been identified as vacancies. In addition, inferring the contribution of a specific type of defect to the general response of the material based on transport or photoresponse measurements might be challenging, as the averaged impact from other defects cannot easily be excluded. Although oxidation has been observed in TMD semiconductors[16] and the absence of IGS due to oxygen substitution has been discussed based on DFT calculations[30], no direct experimental access to the electronic structure of the individual oxygen-related defect has been reported so far. As mentioned, the interpretation of atomically resolved STM images of 2D-TMDs is not straightforward due to the convolution of structural and electronic effects[25], leading to a non-consistent interpretation of the defect type across the current literature[13,16,17,21,22,27,28]. In the case of TEM, the low threshold for electron beam-induced damage in TMDs[23] makes it difficult to identify intrinsic vacancies from new vacancies created by electron beam irradiation[3]. Furthermore, light elements such as C and O contribute only weakly to TEM image contrast. While there is a clear difference between an extant chalcogen and a top or bottom vacancy, the difference between a chalcogen vacancy and an oxygen substitution is quite subtle[51] and could only be resolved in high signal to noise images, which require correspondingly high radiation doses that introduce many new defects. Furthermore, the lack of direct access to the electronic structure of individual defects by TEM hinders further and direct differentiation between defects presenting similar contrast. We further note that the annealing treatment used here extends not only to MBE or CVD grown samples for their characterization in UHV conditions, but also to transferred samples in order to remove contamination caused by air exposure or residues from the transfer process[16–18,36]. Therefore, our conclusions about the prevalence of substitutional oxygen in these 2D-TMDs are expected to be quite general. As the presence of IGS has been connected to key photophysical properties of these materials, identifying the nature of these common defects will advance efforts to control functionality in the emerging 2D-TMD class of systems.

In conclusion, we use experiment and theory to identify substitutional oxygen as a prolific point defect in 2D-$MoSe_2$ and 2D-$WSe_2$, by directly correlating atomic structures and local spectroscopy. We show how the described isolated methods—nc-AFM, STM and STS—could not be used to uniquely determine the structure of the most abundant type of defects in our 2D-$MoSe_2$ and 2D-$WS_2$ samples. Our calculations predict that oxygen substituted in the chalcogen sublattice does not form deep in-gap states, consistent with our STS measurements. Our findings suggest that substitutional oxygen point defects, and not just chalcogen vacancies, have an important role in determining TMD photophysics and guiding current efforts towards increased device functionality.

## Methods

**Experimental details**. Single layers of $MoSe_2$ were grown by molecular beam epitaxy on epitaxial bilayer graphene (BLG) on 6H-SiC(0001). The growth process was the same as described in ref. [52]. The structural quality and the coverage of the $MoSe_2$ samples were characterized by in situ reflection high-energy electron diffraction (RHEED), low-energy electron diffraction (LEED), and photoemission spectroscopy (PES) at the HERS end station of Beamline 10.0.1, Advance Light Source, Lawrence Berkeley National Laboratory. The $WS_2$ films were grown on epitaxial graphene/SiC substrates by a modified chemical vapor deposition process at $T = 900$ °C, which uses $H_2S$ as chalcogen source and $WO_{2.9}$ powder as metal source as described in detail in ref. [53]. The data discussed in the manuscript have been reproducible measured over different sets of $MoSe_2$ and $WS_2$ samples, grown with the described methodology. The samples investigated here were prepared under fundamentally different conditions. Whereas the $MoSe_2$ samples were grown by MBE from elemental sources and experienced UHV conditions in which they were subsequently capped by a protective thin layer of Se, the CVD $WS_2$ samples were grown from a metal oxide precursor and $H_2S$ gas and exposed to air. There are several stages in which the oxygen substitution could be introduced: during the growth itself (particularly for the CVD sample), under ambient conditions (atmospheric $H_2O$, $O_2$, or $CO_2$ could be potential reactants) or while annealing in-vacuum previously adsorbed molecules on vacancy sites that could split and leave the O behind.

STM/nc-AFM imaging and STS measurements were performed at $T = 4.5$ K in a commercial Createc—UHV system equipped with an STM/qPlus sensor. STS differential conductance (d$I$/d$V$) point spectra and spatial maps were measured in constant-height mode using standard lock-in techniques ($f = 775$ Hz, $V_{r.m.s.} = 2.1$ mV, $T = 4.5$ K). d$I$/d$V$ spectra from Au(111) were used as an STS reference to control tip quality. Nc-AFM images were recorded by measuring the frequency shift of the qPlus resonator (sensor frequency $f_0 = 30$ kHz, $Q = 25000$) in constant-height mode with an oscillation amplitude of 180 pm. Nc-AFM images were measured at a sample bias $V_s = -50$ mV, using a tip functionalized with a single CO molecule. STM/STS data were analyzed and rendered using WSxM software[54].

**Theoretical details**. Calculations proceed in two steps. First, we perform DFT calculations within the local density approximation (LDA)[49] for a single vacancy or substitutional defect in a large $5 \times 5$ supercell, corresponding to a small defect concentration of 2%. In this large supercell, which includes 74 atoms and 15 Å of vacuum, the defect density is low enough so that interactions between them can be safely neglected[50]. For the structural relaxation, we constrain the external lattice vectors of the supercell to the experimetnal value and relax the inner structure with LDA, following our previous study[50]. Our DFT-LDA calculations use norm-conserving pseudopotentials, and we explicitly treat of 4 s and 4p semi-core electrons in Mo. (Full details of our DFT-LDA calculations appear in the Supplementary Information) Second, we perform one-shot $G_0W_0$ calculations on our relaxed defect structures, starting from DFT-LDA. To increase accuracy, our GW calculations rely on a fine non-uniform sampling of reciprocal space[55]. Since the defect system contains a mixture of localized and extended states, we explore different treatments of the frequency dependence of the dielectric response in the screened Coulomb interaction, W. We find that the qualitative picture remains the same regardless of whether the frequency dependence is treated in full[56–58], approximated by a Hybertsen-Louie generalized plasmon pole (HL-GPP) model[44,45], or neglected entirely in the static limit. All GW results shown are obtained with the HL-GPP model[44,45]. Further computational details are provided in the Supplementary Information.

## Data availability

The data that support the findings of this study are available in the Supplementary Information and from the corresponding author upon reasonable request.

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

## Acknowledgements

This work was supported by the Center for Computational Study of Excited State Phenomena in Energy Materials (C2SEPEM), which is funded by the U.S. Department of Energy, Office of Science, Basic Energy Sciences, Materials Sciences and Engineering Division under Contract No. DE-AC02-05CH11231, as part of the Computational Materials Sciences Program. Work performed at the Molecular Foundry was also supported by the Office of Science, Office of Basic Energy Sciences, of the U.S. Department of Energy under the same contract number. S.B. acknowledges fellowship support by the European Union under FP7-PEOPLE-2012-IOF-327581. S.B. and M.M.U acknowledge Spanish MINECO (MAT2017-88377-C2-1-R). S.R.A acknowledges Rothschild and Fulbright fellowships. B.S. appreciates support from the Swiss National Science Foundation under project number P2SKP2_171770. A.P. and O.V.Y. acknowledge support by the ERC Starting grant "TopoMat" (Grant No. 306504). M.F.C. acknowledges support from the U.S. National Science Foundation under project number EFMA-1542741. C.H. acknowledges support from NRF grant funded by the Korea government (MSIT) (No. 2018R1A2B6004538). DFT calculations were performed at the Swiss National Supercomputing Centre (CSCS) under project s832 and the facilities of Scientific IT and Application Support Center of EPFL. This research used resources of the National Energy Research Scientific Computing Center (NERSC), a DOE Office of Science User Facility supported by the Office of Science of the U.S. Department of Energy under Contract No. DE-AC02-05CH11231 for the GW calculations. This research used resources of the Advanced Light Source, which is a DOE Office of Science User Facility under contract no. DE-AC02-05CH11231.

## Author contributions

S.B., S.W., M.M.U., O.Y. and A.W.-B. conceived the initial work. S.B., S.W. and A.W.-B. designed the experimental research strategy. S.B., S.W. and B.S. performed the STM/STS/ nc-AFM measurements. A.W.-B. supervised the STM/STS/nc-AFM measurements. S.R.-A., D.Y.Q., S.G.L. and J.B.N. provided the DFT and GW theory support on defects on $MoSe_2$ and $WSe_2$. A.P. and O.Y. provided DFT calculation of defects on $MoSe_2$. H.R. and C.H. performed the MBE growth and characterization of the samples. S.K.M. supervised the MBE growth. C.K., C.C., S.A. and A.S. grew the samples and developed the CVD vdW epitaxy process. M.M.U., M.F.C. and D.F.O. participated in the interpretation of the experimental data. S.B. wrote the manuscript with help from S.R.A., B.S., D.Y.Q., D.F.O., J.B.N. and A.W.-B. All authors contributed to the scientific discussion and manuscript revisions.

## Additional information

**Competing interests:** The authors declare no competing interests.

