## [Peer Review File · Nature Communications]

Reviewers' comments:

Reviewer #1 (Remarks to the Author):

In the paper entitled "Identifying substitutional oxygen as a prolific point defect in monolayer transition metal dichalcogenides with experiment and theory", Sara Barja and coworkers provide a detailed study on the atomic and electronic structure of the "most abundant chalcogen-site point defect common to 2D-TMD semiconductors" (see comment below why I put this in quotation marks). Their careful study using advanced experimental and theoretical techniques shows that in MoSe₂ and WS₂ the common chalcogen defect is actually a substitutional defect, where a chalcogen atom is substituted by an oxygen atom.

The paper is well written, the figures are instructive, the experimental methods as well as the calculations are state of the art and the results are really interesting. I can only recommend publication. I have only one question which should definitely be addressed by the authors:

Why do they not provide the full picture?

The same authors published a work on arxiv which they also cite in the present manuscript as Ref. 39, without a link to the arxiv. This reference is mentioned once:

"By annealing our WS₂ samples at about 600°C in ultra high vacuum (UHV) conditions, we observe the formation of sulfur vacancies. The STS spectra of the VS in WS₂ reveals a characteristic fingerprint with two narrow unoccupied defect states³⁹."

This already shows that other statements they make in the paper cannot be correct. Things like:

"Our results and analysis strongly suggest that the common chalcogen defects in monolayer MoSe₂ and monolayer WS₂, prepared and measured in standard environments, are substitutional defects, where a chalcogen atom is substituted by an oxygen atom, rather than vacancies."

or

"Taken together, our analysis suggests that the most commonly observed point defects in MoSe₂ and WS₂ are O substitutions at Se and S sites, respectively."

cannot be the full picture. I would rather tend to something like: it depends on the preparation method and whether one anneals or not.

Don't get me wrong! I really like the paper. The experiments are state of the art, the calculations definitely took quite some time and both together are really convincing. And finally, together with the paper published on arxiv, someone shows with well-converged calculations that SOC is really important to understand the defect states in TMDs – I was wondering why most studies just looked at the defects without including SOC when I was seeing a huge splitting in my own calculations (yet unpublished).

In my opinion, the authors should either rephrase some of their statements to make clear that it – of course – always depends on the preparation method of the samples, or they should include the part published on arxiv which was surely done on the same samples.

Minor things:

The authors state in the manuscript as well as in supplementary information, that a canceling of tunneling matrix elements due to symmetry reasons (that would prevent them from detecting the in-gap states) has been ruled out, e.g., manuscript page 6 – 2nd paragraph or supplement page 3 – near the end of the long, first paragraph. Yet, they do not provide details how they can rule this out. Please clarify!

In the manuscript as well as in the supplementary information the authors cite a reference by, e.g., "as described in ref. Xx" and then there is a superscript for a different reference.

Reviewer #2 (Remarks to the Author):

Barja et al. combined low-temperature nc AFM, STM/S, and ab initio calculations to determine both the atomic structure and electronic properties of point defects in monolayer MoSe₂ and WS₂. The authors obtained very nice STM and nc AFM image of the point defects, which showed the atomic

structure of the defect quite well. Combined the experimental and calculation results, the author argued that the common defects in monolayer MoSe₂ and WS₂ prepared and measured in standard environments are substitutional oxygen rather than chalcogen vacancies. The correlation between structural defects and electronic properties is essential for understanding and controlling the functionality of 2D-TMDs. The topic of the work is interesting and important. However, I think the author need to provide further evidenced to really support their conclusion.

First, the nc-AFM and STM images of chalcogen defects can be identified to be both chalcogen vacancies and substitutional oxygen. The reason why the author believe that the defect is substitutional oxygen instead of chalcogen vacancies is that they did not observed defect states with the band gap in their STS experiment. I would suggest the author to provide more experimental evidence for their conclusion. For instance, can the author generate some chalcogen vacancy compare their STS results. In some previous work (i.e. Nat. Mater. 2016, 15, 48–53, ACS Nano, 2018, 12,2569), researchers used ion sputtering to generate S vacancy in MoS₂ and observed in gap state in the STS measurement. Since sputtering gun is quite common in STM system, maybe the author can try to gently irradiate their sample and do the STM/S measurement to compare. If it is indeed substitutional oxygen in the samples, the defect density can be increased by exposing to oxygen or decrease by annealing in H₂S (Nature Chemistry,2018,10,1246) . Can author provide some evidence for such changes in their sample?

Secondly, the author believed that in the sample they prepared the dominated defects were substitutional oxygen. They made this conclusion applicable to all the 2D-TMDs. I think the authors need to provide more experimental evidence to show that their samples are really representative for the TMD materials reported in literature. For instance, can they show Raman, PL or transport measurement results of their samples?

Finally, if the conclusion is really applicable to all the TMD materials, the author needs to discuss the reason for this. Is the oxidation of the basal plane is a spontaneous process for all the TMD materials?

Reviewer #3 (Remarks to the Author):

Barja and coworkers report a combination of nc-AFM, STM, and DFT calculations to investigate the nature of defects in two monolayer semiconducting TMDs (MoSe₂ and WS₂). Prior studies of these systems have reported conflicting results, but often conclude that the dominant defect species in these crystals are chalcogen vacancies. In contrast, this study concludes that oxygen substitutions are the primary defects in these materials. The authors motivate that correlating atomic-scale

structural information from nc-AFM, spectroscopy from STS, and comparison to GW-level DFT simulations is critical for distinguishing between chalcogen vacancies and oxygen substitutions, which their work does for the first time.

The manuscript is clearly written and I believe the arguments are well-justified. Despite the relatively straightforward results, I believe the conclusions are important to the community for future materials development and interpretation of experimental results. I therefore recommend publication of this work in Nature Communications. I have only two comments/questions:

1) The caption of Fig. 4a indicates a comparison of the band edges from experiment and theory, however I only see one set of energies in the figure panel. The experimental results appear to be missing.

2) I wish this study had also included comparable results for MoS₂ and WSe₂, which have been the more popular materials in the experimental literature. Are the authors planning to extend this study to those materials as well? At the very least, could comparable DFT simulations be added to the supplement of this work to identify the predicted defect structure for those materials as well? Specifically, do comparable simulations predict the absence of IGS from oxygen substitutions in those materials as well?

Response to Reviewer #1:

In the paper entitled "Identifying substitutional oxygen as a prolific point defect in monolayer transition metal dichalcogenides with experiment and theory", Sara Barja and coworkers provide a detailed study on the atomic and electronic structure of the "most abundant chalcogen-site point defect common to 2D-TMD semiconductors" (see comment below why I put this in quotation marks). Their careful study using advanced experimental and theoretical techniques shows that in MoSe₂ and WS₂ the common chalcogen defect is actually a substitutional defect, where a chalcogen atom is substituted by an oxygen atom.

The paper is well written, the figures are instructive, the experimental methods as well as the calculations are state of the art and the results are really interesting. I can only recommend publication. I have only one question which should definitely be addressed by the authors: Why do they not provide the full picture?

The same authors published a work on arxiv which they also cite in the present manuscript as Ref. 39, without a link to the arxiv (Fix reference). This reference is mentioned once: "By annealing our WS₂ samples at about 600°C in ultra high vacuum (UHV) conditions, we observe the formation of sulfur vacancies. The STS spectra of the V_s in WS₂ reveals a characteristic fingerprint with two narrow unoccupied defect states³⁹. "This already shows that other statements they make in the paper cannot be correct. Things like: "Our results and analysis strongly suggest that the common chalcogen defects in monolayer MoSe₂ and monolayer WS₂, prepared and measured in standard environments, are substitutional defects, where a chalcogen atom is substituted by an oxygen atom, rather than vacancies." Or "Taken together, our analysis suggests that the most commonly observed point defects in MoSe₂ and WS₂ are O substitutions at Se and S sites, respectively." cannot be the full picture. I would rather tend to something like: it depends on the preparation method and whether one anneals or not.

Don't get me wrong! I really like the paper. The experiments are state of the art, the calculations definitely took quite some time and both together are really convincing. And finally, together with the paper published on arxiv, someone shows with well-converged calculations that SOC is really important to understand the defect states in TMDs – I was wondering why most studies just looked at the defects without including SOC when I was seeing a huge splitting in my own calculations (yet unpublished).

In my opinion, the authors should either rephrase some of their statements to make clear that it – of course – always depends on the preparation method of the samples, or they should include the part published on arxiv which was surely done on the same samples.

Response:

The reviewer refers to the split submission of our two recent manuscripts, the current one and "Large spin-orbit splitting of deep in-gap defect states of engineered sulfur vacancies in monolayer WS₂" by Schuler et al., [arXiv:1810.02896](https://arxiv.org/abs/1810.02896) [cond-mat.mtrl-sci]. Although we agree that the two manuscripts complement each other, their specific messages are very distinct. First, the current paper by Barja *et al.*, identifies the most common defect found in our samples as a chalcogen atom substituted by an oxygen atom, rather than a vacancy, as generally discussed in the current literature. Here we provide a comprehensive and direct comparison among some common techniques to study this type of defects at the atomic scale - nc-AFM, STM and STS – and show how neither of those isolated methods could reveal the structure of the most abundant type of defect in

our 2D-MoSe₂ and 2D-WS₂ samples. Second, the paper by Schuler *et al.*, goes beyond the identification of the type defect. It focuses on the so far unobserved electronic signature of S vacancies in 2D-WS₂ and demonstrates how to create actual chalcogen vacancies. To provide the unambiguous insight presented in both papers we established new and independent experimental and theoretical approaches. Hence, based on the above arguments on the scientific insight we provide in each work, we consider there is an urge to separate the messages, albeit their complementarity.

We appreciate the referee's comment and agree that although we observe oxygen substitution in the case of MoSe₂ and WS₂ samples prepared by both MBE (which does not involve any oxygen) and CVD (which involves WO₃ as a precursor), this observation might not be generalizable to all 2D-TMD semiconductors and environmental conditions.

In the manuscript, we argue the general applicability of our results to other TMD samples based on the standardized annealing treatment prior to sample characterization under different environmental conditions (page 10, line 20):

"We further note that the annealing treatment used here extends not only to MBE or CVD grown samples for their characterization in UHV conditions, but also to transferred samples in order to remove contamination caused by air exposure or residues from the transfer process^{16-18,36}. Therefore, our conclusions about the prevalence of substitutional oxygen in these 2D-TMDs are expected to be quite general."

We also note that S vacancies can be deliberately created using non-standard annealing treatments. The deliberate creation of S vacancies by annealing 2D-WS₂ in UHV and observation of the mid-gap states helps to support our assignment that the defects in the sample treated with standard techniques are O-substituents, as we can exclude any experimental artifact that might prevent us from measuring an isolated in-gap state. Relevantly, substituent oxygen defects are still much more abundant in the 2D-WS₂ samples after high-temperature anneal (600C for 30min). Notice that standard annealing treatments for desorption of adsorbants before UHV measurements or residues from transferring processes are normally carried out at lower temperatures, in order to avoid degradation of the 2D-TMD semiconductors.

In the revised manuscript:

- We updated Ref. 39 (current Ref. 40) with the corresponding link to our second (Schuler *et al.*, arXiv:1810.02896 [cond-mat.mtrl-sci]) manuscript.
- Following the reviewer's suggestion, we have reworded the claim, avoiding generalizing the oxygen passivation effect to all TMDs prepared by different methods.

Page 2. Line 10

We changed

"the most abundant chalcogen-site point defect common to 2D-TMD semiconductors"

With

an abundant chalcogen-site point defect common to monolayer MoSe₂ grown by molecular beam epitaxy (MBE) and in monolayer WS₂ grown by chemical vapor deposition (CVD).

Page 2. Line 13

We changed

“the common chalcogen defects in monolayer MoSe₂ and monolayer WS₂, prepared and measured in standard environments, are substitutional defects”

With

the common chalcogen defects in the described 2D-TMD semiconductors, measured in vacuum environment after gentle annealing, are substitutional defects.

Page 9. Line 27.

We changed

“Taken together, our analysis suggests that the most commonly observed point defects in MoSe₂ and WS₂ are O substitutions at Se and S sites, respectively.”

With

Taken together, our analysis suggests that commonly-observed point defects in MoSe₂ and WS₂, growth ex-situ by MBE and CVD and measured in UHV after gentle annealing, are O substitutions at Se and S sites, respectively.

Rev. #1. Minor things:

The authors state in the manuscript as well as in supplementary information, that a canceling of tunneling matrix elements due to symmetry reasons (that would prevent them from detecting the in-gap states) has been ruled out, e.g., manuscript page 6 – 2nd paragraph or supplement page 3 – near the end of the long, first paragraph. Yet, they do not provide details how they can rule this out. Please clarify!

In the manuscript as well as in the supplementary information the authors cite a reference by, e.g., "as described in ref. Xx" and then there is a superscript for a different reference.

Response:

The wavefunctions composing the tip- and defect-related states might be orthogonal leading to a vanishing tunneling matrix element. This scenario can be discarded from experimental evidence as the defect was probed with metallic and CO functionalized tips that were shown to exhibit s- and p-wave character, none of which detected an in-gap state.

In the revised manuscript:

- We included the former argument in page 3, line 16 of the supplementary information. **We considered the case that the wavefunctions composing the tip- and defect-related states might be orthogonal, leading to a vanishing tunneling matrix element and vanishing current related to these states. However, this scenario is unlikely based on the experimental evidence at hand: the defect was probed with metallic and CO-functionalized tips, which have been shown to exhibit states having s- and p-wave character. Neither of these two orbital symmetries detected an in-gap state.**

- We corrected
Ref. 25²⁵ (main manuscript, page 6, line 3),
Ref. 52⁵² (main manuscript, page 11, line 14),
Ref. 53⁵³ (main manuscript, page 11, line 20)
and **Ref. 15**¹⁵ (supplementary information, page 7, line 14)

to fit the superscript reference.

Response to Reviewer #2:

Barja et al. combined low-temperature nc AFM, STM/S, and ab initio calculations to determine both the atomic structure and electronic properties of point defects in monolayer MoSe2 and WS2. The authors obtained very nice STM and nc AFM image of the point defects, which showed the atomic structure of the defect quite well. Combined the experimental and calculation results, the author argued that the common defects in monolayer MoSe2 and WS2 prepared and measured in standard environments are substitutional oxygen rather than chalcogen vacancies. The correlation between structural defects and electronic properties is essential for understanding and controlling the functionality of 2D-TMDs. The topic of the work is interesting and important. However, I think the author need to provide further evidenced to really support their conclusion.

First, the nc-AFM and STM images of chalcogen defects can be identified to be both chalcogen vacancies and substitutional oxygen. The reason why the author believe that the defect is substitutional oxygen instead of chalcogen vacancies is that they did not observed defect states with the band gap in their STS experiment. I would suggest the author to provide more experimental evidence for their conclusion. For instance, can the author generate some chalcogen vacancy compare their STS results. In some previous work (i.e. Nat. Mater. 2016, 15, 48–53, ACS Nano, 2018, 12,2569), researchers used ion sputtering to generate S vacancy in MoS₂ and observed in gap state in the STS measurement. Since sputtering gun is quite common in STM system, maybe the author can try to gently irradiate their sample and do the STM/S measurement to compare. If it is indeed substitutional oxygen in the samples, the defect density can be increased by exposing to oxygen or decrease by annealing in H₂S (Nature Chemistry,2018,10,1246) . Can author provide some evidence for such changes in their sample?

Response:

The Reviewer is asking to provide further evidence for the identification of the defects as substitutional oxygen instead of chalcogen vacancies, by generating chalcogen vacancies and comparing the STS spectra.

In page 7, line 10 we describe: "By annealing our WS₂ samples at about 600°C in ultra high vacuum (UHV) conditions, we observe the formation of sulfur vacancies. The STS spectra of the VS in WS₂ reveals a characteristic fingerprint with two narrow unoccupied defect states³⁹." The deliberate creation of S vacancies by annealing 2D-WS₂ in UHV and observation of the mid-gap states support our assignment as O-substituents, as we can exclude any experimental artifact that might restrict the in-gap measurement.

In addition, we discussed the potential origin of oxygen incorporation in page 6, line 20:

“Based on growth conditions for both single layers (MoSe₂ and WS₂) and the sample treatment prior to our scanning probe measurements, oxygen, carbon, silicon, nitrogen, and hydrogen may potentially substitute chalcogen atoms. We can exclude S and Se substituents in, respectively, MoSe₂ and WS₂ samples, as they are grown in different experimental set-ups, avoiding cross contamination of chalcogen sources”

page 6, line 28:

“Oxygen substitution, on the other hand, is predicted to suppress the deep in-gap states associated with the vacancies^{19,29}, and oxygen is present during the CVD growth process of WS₂. While the MBE samples are Se-capped to prevent air exposure during the transfer to the STM set-up, desorption of atmospheric H₂O, O₂, or CO₂ from the air-exposed sample holder is likely to occur during the annealing process for decapping.”

and page 11, line 23:

“Whereas the MoSe₂ samples were grown by MBE from elemental sources and experienced UHV conditions in which they were subsequently capped by a protective thin layer of Se, the CVD WS₂ samples were grown from a metal oxide precursor and H₂S gas and exposed to air. There are several stages in which the oxygen substitution could be introduced: during the growth itself (particularly for the CVD sample), under ambient conditions (atmospheric H₂O, O₂, or CO₂ could be potential reactants) or while annealing in vacuum previously adsorbed molecules on vacancy sites that could split and leave the O behind.”

In the revised manuscript:

- We updated Ref. 39 (current Ref. 40) with the corresponding link to our second (Schuler et al., arXiv:1810.02896 [cond-mat.mtrl-sci]) manuscript, so the reader can verify the emergence of the deep-in gap state associated with the chalcogen vacancy. (See response to Reviewer #1)
- We included the work in the *ACS Nano*, 2018, 12,2569 (current Ref. 17), as referred by the reviewer.

Rev. #2. *Secondly, the author believed that in the sample they prepared the dominated defects were substitutional oxygen. They made this conclusion applicable to all the 2D-TMDs. I think the authors need to provide more experimental evidence to show that their samples are really representative for the TMD materials reported in literature. For instance, can they show Raman, PL or transport measurement results of their samples?*

Finally, if the conclusion is really applicable to all the TMD materials, the author needs to discuss the reason for this. Is the oxidation of the basal plane is a spontaneous process for all the TMD materials?

Response (also in response to Reviewer #1):

We thank the referees for their appreciation of the main message in our paper. As discussed in the response to Reviewer #1, although oxygen substitution is demonstrated in the case of MoSe₂ and WS₂ samples prepared by MBE (which does not involve any oxygen) and CVD (which involves WO₃ as precursor), respectively, this might not be representative for other TMD materials or under different environmental condition.

In the manuscript, we discuss the potential generalization of our results to other samples based on the standardized annealing treatment prior to sample characterization under different environmental conditions.

Page 10. Line 20:

“We further note that the annealing treatment used here extends not only to MBE or CVD grown samples for their characterization in UHV conditions, but also to transferred samples in order to remove contamination caused by air exposure or residues from the transfer process^{16-18,36}. Therefore, our conclusions about the prevalence of substitutional oxygen in these 2D-TMDs are expected to be quite general.”

In the revised manuscript we have reworded the claim, avoiding generalizing the oxygen passivation effect to all TMDs prepared by different methods.

Page 2. Line 10

We changed

“the most abundant chalcogen-site point defect common to 2D-TMD semiconductors”

With

an abundant chalcogen-site point defect common to monolayer MoSe₂ grown by molecular beam epitaxy (MBE) and in monolayer WS₂ grown by chemical vapor deposition (CVD).

Page 2. Line 13

We changed

“the common chalcogen defects in monolayer MoSe₂ and monolayer WS₂, prepared and measured in standard environments, are substitutional defects”

With

the common chalcogen defects in the described 2D-TMD semiconductors, measured in vacuum environment after gentle annealing, are substitutional defects.

Page 9. Line 27.

We changed

“Taken together, our analysis suggests that the most commonly observed point defects in MoSe₂ and WS₂ are O substitutions at Se and S sites, respectively.”

With

Taken together, our analysis suggests that commonly-observed point defects in MoSe₂ and WS₂, growth ex-situ by MBE and CVD and measured in UHV after gentle annealing, are O substitutions at Se and S sites, respectively.

Finally, in order to support the former claim, the referee suggests to show Raman, PL or transport measurement results of our samples. For the optical and electronic properties of the MBE-grown MoSe₂ and CVD-grown WS₂ monolayer films on graphene, we refer to M. M. Ugeda, *et al.* Nat. Mat. **13**, 1091 (2014) - Ref. 25 in the main manuscript- and C. Kastl, *et al.* 2018 2D Mater **5**, 045010 (2018) - Ref. 53 in the main manuscript-, respectively, and the corresponding supplementary information. The electronic band structure of our MoSe₂ and WS₂ monolayers on graphene measured by ARPES is in-line with theoretically expected values for quasi-freestanding WS₂. The photoluminescence and Raman signatures of both monolayers agree well with those reported in other studies.

Response to Reviewer #3:

Barja and coworkers report a combination of nc-AFM, STM, and DFT calculations to investigate the nature of defects in two monolayer semiconducting TMDs (MoSe₂ and WS₂). Prior studies of these systems have reported conflicting results, but often conclude that the dominant defect species in these crystals are chalcogen vacancies. In contrast, this study concludes that oxygen substitutions are the primary defects in these materials. The authors motivate that correlating atomic-scale structural information from nc-AFM, spectroscopy from STS, and comparison to GW-level DFT simulations is critical for distinguishing between chalcogen vacancies and oxygen substitutions, which their work does for the first time.

The manuscript is clearly written and I believe the arguments are well-justified. Despite the relatively straightforward results, I believe the conclusions are important to the community for future materials development and interpretation of experimental results. I therefore recommend publication of this work in Nature Communications. I have only two comments/questions:

Rev. #3. Comment 1)

1) The caption of Fig. 4a indicates a comparison of the band edges from experiment and theory, however I only see one set of energies in the figure panel. The experimental results appear to be missing.

Response:

Fig. 4a is composed of three different panels. The left panel, labeled O_{Se}(Exp.), represents the band extrema energies extracted from the experimental dI/dV spectra in Fig. 2. The center panel, labeled V_{Se}(GW), contains the corresponding theoretical energies for the bare Se vacancy calculated using the GW approximation. The right panel, labeled O_{Se}(GW), contains the theoretically calculated energies for the O substitution at a Se site. Based on the reviewer's comment, we have clarified the figure caption and made the figure labels more explicit.

In the revised manuscript

- We modified the labels in Fig. 4a to be “Experiment” (in grey) and “Theory, $V_{\text{Se}}(\text{GW})$ and $\text{O}_{\text{Se}}(\text{GW})$ ” (in black), instead of the former $\text{O}_{\text{Se}}(\text{Exp.})$, $V_{\text{Se}}(\text{GW})$ and $\text{O}_{\text{Se}}(\text{GW})$ bottom labels, and moved the labels from the bottom to the top of the figure. We accordingly modified the line color (now in grey) of the band extrema extracted from the experimental dI/dV .
- We modified the description on the figure caption:

a, Band extrema energies extracted from the experimental dI/dV spectra in Fig. 2 (Experiment, grey) are compared to the corresponding energies (Theory, black) calculated using the GW approximation for the bare Se vacancy $-V_{\text{Se}}(\text{GW})-$ and a substitutional O at a Se site $-\text{O}_{\text{Se}}(\text{GW})-$. Energies of valence (VB) and conduction (CB) bands are indicated by black lines; while the deep in-gap states (IGS) appearing in the V_{Se} is indicated by the red line, which dictates the Fermi level of the calculated system. To aid comparison, all VB energies have been set to zero.

Figure 4. Comparison of the band energy diagrams and the local density of states of pristine and O substitution Se defects in 2D-MoSe₂. a, Band extrema energies extracted from the experimental dI/dV spectra in Fig. 2 (Experiment, grey) are compared to the corresponding energies (Theory, black) calculated using the GW approximation for the bare Se vacancy $-V_{\text{Se}}(\text{GW})-$ and a substitutional O at a Se site $-\text{O}_{\text{Se}}(\text{GW})-$. Energies of valence (VB) and conduction (CB) bands are indicated by black lines; while the deep in-gap states (IGS) appearing in the V_{Se} is indicated by the red line, which dictates the Fermi level of the calculated system. To aid comparison, all VB energies have been set to zero. The three-fold symmetry and spatial extent observed in the experimental dI/dV constant-height conductance map measured at the (b) CB energy ($V_{\text{sample}}=0.7$ V) are seen on both (c) the pristine V_{Se} and (d) the O_{Se} . Similarly, the experimental spatial extent at the (e) VB ($V_{\text{sample}}=-1.5$ V) also reproduced the simulated LDOS of both (f) V_{Se} and (g) O_{Se} .

Rev. #3. Comment 2)

2) I wish this study had also included comparable results for MoS₂ and WSe₂, which have been the more popular materials in the experimental literature. Are the authors planning to extend this study to those materials as well? At the very least, could comparable DFT simulations be added to the supplement of this work to identify the predicted defect structure for those materials as well? Specifically, do comparable simulations predict the absence of IGS from oxygen substitutions in those materials as well?

Response:

The effect of point defects on the DFT bandstructure was previously examined for MoS₂ and WSe₂, as well as for MoSe₂ and WS₂, see, e.g., in Haldar et al., PRB 92, 235408 (2015). While oxygen substitution is not explicitly examined in the mentioned reference, it is evident from it that the effect of point defects on the energy levels is very similar between MoSe₂ and MoS₂, and between WSe₂ and WS₂, as can be expected. In our work we chose to calculate the systems examined in the performed experiments for direct comparison; however, we assume that while the energy gap values vary with different components, the main observations of in-gap states suppression and changes in the relative energy levels alignment are strongly related to the defect type, and not the specific components composing the TMDs. This assumption is further supported by former DFT computations predicting suppression of the deep in-gap states due to oxygen substitution in MoS₂ (*Phys. Rev. Lett.* **109**, 035503 (2012) and *Phys. Chem. Chem. Phys.* **18**, 14001–14006 (2016)), pointing towards a general behavior of absence of IGS from oxygen substitution in 2D-TMD semiconductors.

In the revised manuscript (page 6, line 28) we explicitly refer to the work in references 19 and 29 as DFT predictions in MoS₂.

Oxygen substitution, on the other hand, is predicted to suppress the deep in-gap states associated with the vacancies in MoS₂^{19,29}.

REVIEWERS' COMMENTS:

Reviewer #1 (Remarks to the Author):

The authors have addressed all questions of the referees and thus I can only recommend publication.

Reviewer #2 (Remarks to the Author):

The authors have addressed all the concerns I have. The manuscript can be accepted for publication.

Reviewer #3 (Remarks to the Author):

The updated manuscript satisfactorily addresses my comments, as well as those raised by the other reviewers. I recommend publication of this work.